# Development of Custom Anatomic Healing Abutment Based on Cone-Beam Computer Tomography Measurement on Human Teeth Cross-Section

**DOI:** 10.3390/ma14164570

**Published:** 2021-08-14

**Authors:** Monika Teślak, Adam Ziemlewski, Igor Foltyn, Iwona Ordyniec-Kwaśnica, Barbara Drogoszewska

**Affiliations:** 1Department of Prosthodontics, Medical University of Gdansk, ul.E.Orzeszkowej 18, 80-208 Gdańsk, Poland; iwona.ordyniec-kwasnica@gumed.edu.pl; 2Private Practice Impladent, ul.Kartuska 312, 80-125 Gdańsk, Poland; adam@ziemlewski.pl; 3Private Practice Projekt Uśmiech, ul.Partyzantów 76, 80-214 Gdańsk, Poland; foltyn.igor@hotmail.com; 4Department of Maxillofacial Surgery, Medical University of Gdansk, ul. Smoluchowskiego 17, 80-214 Gdańsk, Poland; barbara.drogoszewska@gumed.edu.pl

**Keywords:** implant, healing abutment, zirconium, titanium

## Abstract

Introduction: Nowadays, the final success of implantation is not only based on obtaining osseointegration of the implant but is also determined by achieving a satisfactory aesthetic effect of the soft tissues surrounding the implant, which can be defined as an aesthetic integration. The process of obtaining this aesthetic integration already begins at the stage of placing the healing abutment, which allows us to obtain the emergence profile necessary for our prosthetic reconstruction. Materials and Methods: The study used cone-beam computer tomography (CBCT) scans of 51 patients. The measurements of the maxillary teeth (central incisor, lateral incisor, canine, first premolar, and first molar) were performed from cross-sections of the individual teeth at the transition zone to design a custom anatomic healing abutment milled from zirconium and luted to the non-index Ti-base. Results: The obtained results allowed to design and create the shape of the anatomic healing abutment. Conclusions: The use of laboratory-produced anatomical healing abutments is possible and may allow to obtain the desired and planned emergence profiles of prosthetic restorations. In addition, it might be a method of reducing work time at the dental chair but further clinical trials are necessary.

## 1. Introduction

One of the greatest challenges of modern implantology is to achieve beautiful and natural aesthetic results. The key to achieving this is the maximum preservation or reconstruction of hard and soft tissues around the implant.

Current guidelines for successful therapeutic implantology treatment are based on the evaluation of the implant and restoration survival, dento-gingival aesthetics, rate of mechanical complications, and the bone levels and health of surrounding soft tissues [1,2,3]. The tissues surrounding the implant differ from the tissues surrounding the teeth by having no periodontium around them and the architecture of the bone surrounding the implant does not have any interdental septum that are subject to resorption after tooth extraction. As a result of the disappearance of hard tissues, the interdental papilla is also lost [4]. The peri-implant tissues are comparable to the structures around the natural dentition, which consist of the connecting epithelium and connective tissue beneath it. The size of the total connection here is about 1–1.5 mm (constant value), while the epithelium connecting around the implants is 1.5–2 mm in the vertical dimension [5,6,7] and connects to the surface of the implant via hemidesmosomes [8]. The peri-implant fiber of the connective tissue runs parallel to the connector or implant surface [5,9] and there are relatively few blood vessels in it, which is one of the reasons for reduced resistance to mechanical and microbiological stimuli [10]. In the case of implants, there is also no anatomic structure comparable to the periodontal fissure [11]. The mucosa around the implant-peri-implant mucosa (PIM-Lindhe et al., 2008) develops within 4–6 weeks after the application of the healing abutment or connector [12]. During the healing process, a transmucosal attachment is created: an attachment of soft tissues connecting to the implant and providing a barrier between the oral environment and bone tissue. For therapeutic success, it is important to reduce bone loss around the implant. The cause of bone resorption around the implant is not from strain but from the exposure to the oral environment, as it occurs regardless of whether the implant is tightened with a connector, crown, or healing abutment [4]. The position of the implant, which determines the long-term healing success, also plays a key role. The guideline for positioning the implant is the prof. Rojas-Vizcaya principle (3 mm apical and 2 mm buccal-principle 3A-2B). The implant should be placed 3 mm above the neck of the planned prosthetic crown to maintain the appropriate biological width and 2 mm from the buccal side should be ensured to compensate for the marginal resorption [13,14]. The position of the implant and prosthetic also affects the presence of the interdental papilla. According to the studies by Priest and Lai et al., the height of the papilla between a single implant and the adjacent teeth depends primarily on the position of the clinical attachment of the adjacent tooth [15]. After attaching the crown on a single implant, the maximum height of the papilla, measured from the top of the bone to the tip of the papilla, may be 4.5 mm, which means that by making the final crown, the tangent point must be placed no further than 4.5 mm from the edge of the bone [5,15,16]. According to the studies conducted by Tarnow et al., in the case of two implants, the maximum height of the papilla is 3.5 mm. The tangent point should therefore be placed 3 mm from the edge of the bone [17].

The soft tissues at the implant have the characteristics of scar tissue, which is the cause of reduced resistance to damaging factors such as the inflammation due to bacterial plaque. This characteristic feature makes it very important to produce the corresponding anatomical structure of the soft tissue around the implant as well as the corresponding shape of crowns and bridges capable of maintaining a high standard of oral cavity hygiene [18]. This procedure will minimize the incidence of inflammation of soft and hard tissues holding the implant.

Prefabricated standard healing abutments have a circular cross-section. They are produced in various sizes and heights, and are usually made of titanium [19,20]. The round cross-section of the screw allows it to be screwed to the implant in any position. The base of the connector compared to the root outline is round and smaller, which results from the fact that it has a maximum diameter of the degree of the implant. From this degree to the emergence of the abutment from the soft tissues, the abutment should have the three-dimensional shape of the reconstructed tooth, which is called the emergence profile [21]. By shaping the emergence profile using the healing abutment or temporary prosthetic placed on the implant abutment, a shape corresponding to the final future crown should be achieved. This is crucial because when the healing process is completed, modifying the shape of the abutment and the prosthetic work that follows may influence a change of the soft tissue around the implant [21].

The healing response around abutments can be evaluated and the soft tissue around the fixtures can heal according to the contours of the definitive prosthesis [22]. The circular shape of the stock-healing abutment makes it more unpredictable in molding the tissue with contours similar to those of natural teeth [23]. To obtain a satisfactory aesthetic effect of soft tissues around the implant, a healing abutment in a shape as close as possible to the future prosthetic crown should be used [20]. When conventional stock-healing abutments are used, the surrounding soft tissue profile at the time of the restorative treatment may be unfavourable, therefore requiring additional time-consuming recontouring [24]. There is a series of publications that provide the evidence of the advantages of customized healing abutments [22,23,25,26]. All of them are fabricated by customized computer aided design and computer aided manufacturing (CAD/CAM) techniques digitally designed for individual patients. However, this technique is time-consuming and requires additional diagnostic and laboratory work. Another way to create an emergence profile is through a dynamic compression with the use of a series of the provisional crowns [27,28]. This technique is favoured for esthetic region as it allows patients to receive a temporary crown. However, this approach requires time and creates an additional cost. The advantages of the individual healing abutment led us to seek a way to obtain an emergence profile corresponding to the natural dentition using a prefabricated anatomical healing abutment and to verify the possibility to develop a custom anatomical healing abutment without additional time-consuming chairside procedures. This design is novel and hopefully, after clinical trials, might be used in everyday implantology. The aim of this study was to design and fabricate an individual anatomy reflecting healing abutment based on cone-beam computed tomography (CBCT) cross-sectioning dedicated to the implant’s location in the upper jaw.

## 2. Materials and Methods

In this study, randomly selected CBCT scans of patients from the database were used regardless of the gender and age of the patient. In the research group, CBCT scans were from 30 women aged from 19 to 65 years and 21 men aged 18 to 69 years, and were randomly selected by two independent researchers. The scans of patients with tooth shape disorders, numerous fillings, and metal scattering problems were disqualified. The scans were originally made for diagnostic purposes due to planned dental treatment. The study had the consent of the bioethical commission at the Regional Medical Chamber in Gdańsk (registration number KB-13/18 approved 25May 2018). The commission operates in accordance with the principles of good clinical practice.

To analyze anatomic crown ratios, the measurements were taken from CBCT scans of the maxillary teeth: central incisor, lateral incisor, canine, first premolar, and first molar made on cross-sections of individual teeth at the place where the tooth comes from bone tissue into the soft tissue (transition zone). Four measurements were made in millimetres on each cross-section (specified in the contract study L, C1, H, and C2). Two independent researchers made the measurements (Figure 1 and Figure 2). The L1 value was measured at the two most convex opposing points in the sagittal plane of the examined tooth and the H value was measured at the two most convex opposing points in the frontal plane of the examined tooth. The C1 value was measured at the two most convex opposing mesially points halfway between lines L and H, and the C2 value was measured at the two most convex opposing distally points halfway between lines L and H (Figure 1 and Figure 2). The obtained results were subjected to a statistic analysis. Due to the fact that all the variables had normal distributions, it was possible to obtain the arithmetic mean without deleting data. Received results allowed for creating an anatomic healing abutment shape in dental CAD/CAM software (Zirkonzahn, Modellier).Z Each abutment was designed at 4 mm tall. The customized healing abutments STL was exported and milled from zirconium block. The milled portion was luted to the non-index titanium CAD/CAM base (Ti-base) with light-cured resin and polished.

## 3. Results

At total of 51 patients were included in this study. Out of 51 patients, 59% (30) were females and 41% (21) were males (Figure 3).

The received results allowed to design the shape of the individualized anatomical healing abutment for each group of the teeth mentioned (Table 1). The arithmetic average for central incisor measurements were L = 6.633 mm, C1 = 6.302 mm, H = 5.889 mm, and C2 = 6.787 mm. The major difference concerning the lateral incisor measurements was the H diameter, which was 1.442 mm less for lateral incisor. The outcome of this distinction is a narrower shape for the customized healing abutment for the lateral incisor. The H diameter for the first premolar (H = 4.569 mm) was similar to the lateral incisor (H = 4.447 mm) with the higher values of other parameters (L, C1, and C2). Such dimensions create the biscuit shape of the first premolar, which is similar to the natural anatomy of this tooth. The measurements for first molar allowed to design a square shape healing abutment. Each healing abutment was created in dental CAD/CAM software (Zirkonzahn, Modellier) based on the results of the CBCT measurement analysis (Figure 4). The access screw hole was situated in a 60% to 40% ratio in the vestibule palatial axis due to the fact that implant placement usually is situated more towards the palate. The customized healing abutments’ STL was exported and milled from zirconium block. The milled portion was luted to the non-index titanium CAD/CAM base (Ti-base) with light-cured resin and polished. The received elements are presented in Figure 5 (Figure 5).

## 4. Discussion

In order to achieve success in implantology treatment, it is highly beneficial to understand the importance of proper designing and the obtaining of the emergence profile. The aim of this study was to design an anatomically stock-healing abutment. We can manage the healing of tissues around the implant by using customized prosthetic elements such as healing abutments or connectors between the implant and the crown. Due to the use of such solutions, the tissue healing process is possible to plan and is more predictable, which results in a better aesthetic effect [1,2,3].

When striving to obtain the best fitting natural representation, we should come as close as possible to the physiological anatomy of the dental and surrounding structures. The use of standard prefabricated implant prostheses such as the healing abutment to shape the emergence profile is an unpredictable procedure. These screws are made in a shape of which the cross-section is circular: in no way does this shape mimic the natural anatomy of the crown and the root of the tooth [29,30], which should ensure us to obtain aesthetic integration. Janakievski’s study [31] reports that because of the round profile of standard healing screws, it is not possible to support supra-crestal soft tissues surrounding the implant. There are techniques based on customizing a pre-manufactured healing screw, e.g., with a composite material placed directly in the patient’s mouth [22] or by a CAD/CAM system after a standard silicone impression or an intra-oral scan and CBCT planning [19,20]. These techniques enjoy good results and increasing popularity, although unfortunately they carry the need for additional equipment such as CAD/CAM and require the dentist to spend extra time with the patient and often arrange additional visits, which significantly extends the treatment period.

The use of existing CBCT scans taken from the research group allowed us to obtain a cross-section of dimensions needed to obtain average values in four dimensions (L, C1, H, and C2) at the entry of the tooth into the bone structure (transition zone) for the central incisor, lateral incisor, canine, first premolar, and first molar in the upper jaw. The obtained results allowed to design the shape of the individualized anatomical healing abutment for each group of teeth mentioned. None of the profiles received were of a rounded profile due to the detailed measurements, making it feasible to get as close to the natural anatomy as possible. Having such an anatomically shaped healing abutment offers the possibility to control the process of soft tissue healing around the implant either immediately after implantation or after the integration period and after unveiling the implant without the necessity to establish additional visits with the patient or long procedures involving the use of additional materials needed for standard healing abutment

The material from which the implant and prosthetic components are made should be mechanically stable (abutments) and maximally biocompatible (abutments and healing abutments). In the studies by Abrahamsson et al. performed on animals, materials for making abutments were tested [32]. The abutments made of titanium and zirconium gave satisfactory results in the form of creating a biological width similar to that of natural teeth, whereas gold alloy and dental porcelain abutments resulted in increased peri-implant bone loss and soft tissue recession, with the highest value of use attributed the last one [13].

While selecting material for customized implant abutments, recent advances in milling technology recommend two materials: zirconium and titanium [3]. Titanium has very good material properties and high biocompatibility, although it is also fairly expensive and does not allow for an easy modification of the shape [33,34]. Titanium abutments also create a dark-colour shine through soft tissues, which is an unsightly defect unacceptable by various patients [35].

The meta-analysis by Linkevicus et al. showed statistically significant superiority of Zr abutments over Ti abutments in developing natural soft tissue colour and a superior aesthetic PES score [3]. When designing a healing abutment, the focus should mainly be on biocompatibility, ease of shape modification, and economy, hence the choice of using zirconium. The selection of zirconia ceramic for healing abutments also arises from the low bacterial adherence to the zirconia surface as shown in different in vitro and in vivo studies [36,37,38].

Nowadays, this material is also relatively inexpensive, thanks to which it is possible to create a highly biocompatible and economic implant-prosthetic solution that is an individualized anatomical pre-fabricated healing abutment.

The present study is a preliminary report on an innovative method of using clinical data for the construction of anatomical healing abutments made of biocompatible material, namely zirconium and available chairside without additional time-consuming visits or protocols. Further clinical trials, which are currently underway, are needed to determine the results of the study and compare its effectiveness with other solutions practiced so far.

## 5. Conclusions

The use of dental implants in the treatment of missing teeth is a predictable treatment at the osseointegration stage. The use of individualized healing abutment allows to achieve this predictability in the healing process of soft tissues surrounding the implant and in the formation of the emergence profile. The obtained CBCT measurement results allowed to design and create the shape of the anatomic healing abutment, milled from biocompatible zirconium and luted to the non-hexed Ti-base.

The use of laboratory-produced anatomical healing abutments is possible and may allow to obtain the desired and planned emergence profile of prosthetic restoration. It might also be a method of reducing work time at the dental chair but further clinical trials are necessary.

## Figures and Tables

**Figure 1 materials-14-04570-f001:**
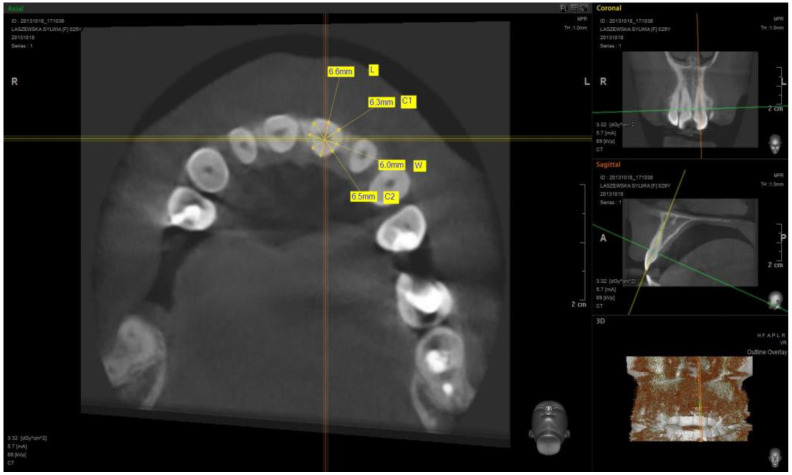
Example of a measurement of the central incisor cross-section.

**Figure 2 materials-14-04570-f002:**
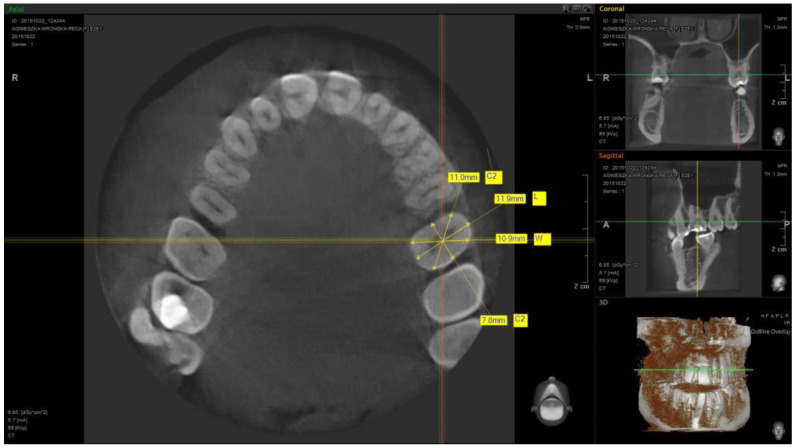
Example of a measurement of the first molar cross-section.

**Figure 3 materials-14-04570-f003:**
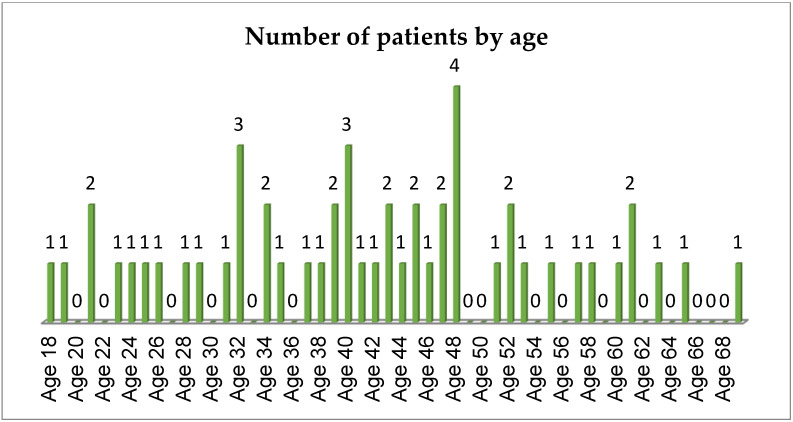
The number of patients by age.

**Figure 4 materials-14-04570-f004:**
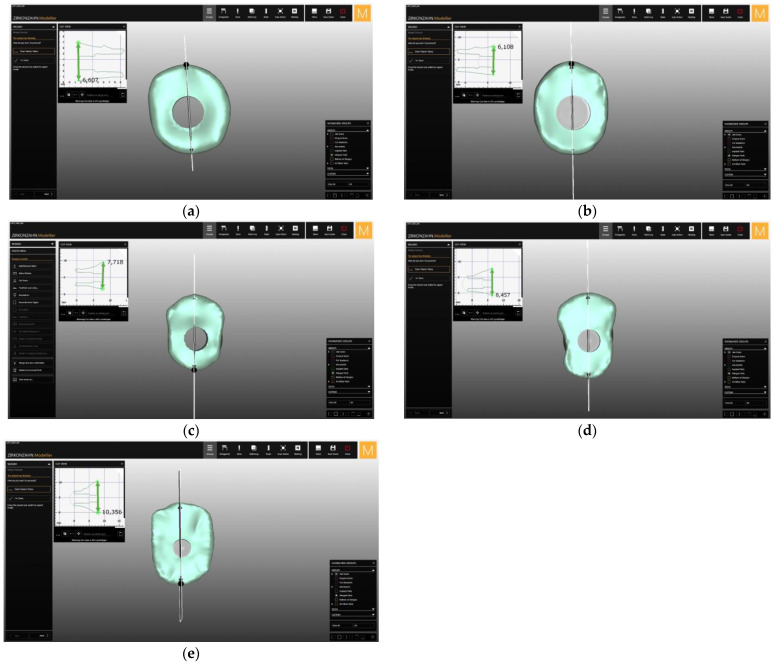
The laboratory design of the healing abutment for the (**a**) central incisor, (**b**) lateral incisor, (**c**) canine, (**d**) first premolar, and (**e**) first molar in the upper jaw.

**Figure 5 materials-14-04570-f005:**
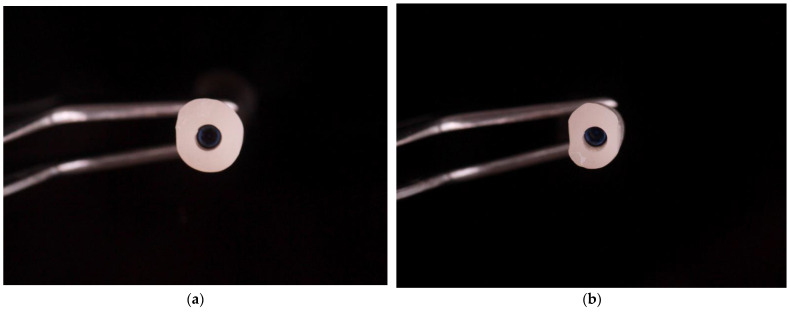
Milled zirconium anatomical healing abutment for the (**a**) central incisor, (**b**) lateral incisor, (**c**) canine, (**d**) first premolar, and (**e**) first molar in upper jaw.

**Table 1 materials-14-04570-t001:** Results of the measurements (mm) of L, C1, H, and C2 for the central incisor, lateral incisor, canine, first premolar, and first molar (mm).

	Central Incisor				Lateral Incisor				Canine					First Premolar				First Molar		
	L	C1	H	C2	L	C1	H	C2	L	C1	H	C2	L	C1	H	C2	L	C1	H	C2
**Arithemic Average**	6.633	6.302	5.889	6.787	6.085	5.65	4.447	5.533	7.693	7.023	5.358	6.536	8.509	7.611	4.569	6.957	10.320	9.963	7.577	10.084
**Standard Deviation**	0.546	0.478	0.600	0.538	0.525	0.545	0.579	0.506	0.578	0.599	0.532	0.679	0.619	0.634	0.602	0.704	0.751	1.023	0.696	1.159
**Median**	6.6	6.2	5.9	6.8	6.1	5.7	4.4	5.45	7.7	7	5.4	6.45	8.45	7.8	4.55	7	10.35	10.1	7.6	10.3
**Minimum Value**	5.8	5.5	4.5	5.5	5.1	4.4	3.4	4.5	6.4	5.9	4.5	5.2	7.2	6.2	3.1	5	8.1	7.3	6.4	7.1
**Maximum Value**	8	7.5	7.7	8.6	7.4	7	6.4	6.9	8.9	8.6	6.6	8.2	9.8	8.7	5.8	8.3	11.8	11.9	9	12.4

## Data Availability

The data presented in this study are available upon request from the corresponding author.

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
