# Peer review of "Development of Custom Anatomic Healing Abutment Based on Cone-Beam Computer Tomography Measurement on Human Teeth Cross-Section"

_materials, 2021, doi:10.3390/ma14164570_

Round 1
Reviewer 1 Report
Dear authors:
The topic analyzed is interesting.
You describe a new technique to design the shape of the healing abutment. However, there is no clinical evaluation of the emergence profile obtained with these healing abutments. Furthermore, conclusions are based on assumptions on possible clinical effects, but there is no clinical evidence of it.
Moreover, there are several criticisms regarding research problems, methodology, or discussion. Please see list below.
-The writing should be improved. There are some errata in the text that need edit. Please review the manuscript.
Introduction:
-Please describe why this study is novel and different from what was previously published
- There is no hypothesis in the proposed study. Please introduce the study hypothesis.
Material and Methods:
-This section needs improvement and further detail.
-Please discuss how the authors determined that sample size was adequate. Was a power analysis performed?
-Please explain the statistical analysis performed.
Results
The authors say: “Each healing abutment was created in dental CAD/CAM software 158 ……The milled portion was luted 162 to the non-index titanium CAD/CAM base (Ti-base) with light-cured resin and polished”. This paragraph must be included in the Materials and Methods section
Discussion
-Please describe the impact of the study results on clinical practice
-Introduce the limitations of the study in the Discussion section.
Best regards
Author Response
Please see the attachment.
Regards,
Authors

Reviewer 2 Report
Figure 5 is not reflected in the text (or at least I can't find it)
I do not observe errors in bibliography numbering.
Also not in Tables, although table 1 should try to edit in HORIZONTAL.
Various aspects could be emphasized in the discussion:
Has this technique been designed for immediate post-extraction implants (personalized abutment) or could it be used for any type of posterior placement in this patient?
In this case, the changes that occur in the post-extraction socket should be discussed since the situation in fully healed bone can vary greatly (and of course also in soft tissues).
The article, although limited, could be published with these minor corrections.
Author Response

(The authors gave the same response as above.)

Reviewer 3 Report
The study of “Development of Custom Anatomic Healing Abutment Based on Cone-Beam-Computer-Tomography Measurement on Human Teeth Cross Section” used cone beam computed tomography (CBCT) images to measure the angulation of teeth including central incisor, lateral incisor, canine, first premolar and first molar.
Overall, it is a clinical orientated study. However, there are some major problems needed to be fixed before submission. English writing and the structure of the manuscript are required to be improved also.
- The abstract is required to be rewritten. The abstract should include the information related to objective, materials and methods, results and conclusion.
- The matters of introduction section and discussion section are not easy to be understood. English writing and the structure of those two parts are required to be improved.
- For example: CBCT, the full name should be written where the abbreviation first appears, rather than showing it later. This makes it difficult for readers to read.
- For section of Materials and methods, what are the inclusion and exclusion criteria for the data selection?
- The description about the method is vague.
A. It is not easy to understand how the angulation of tooth is measured.
B. For the Figure 1 and Figure 2, the words and arrows inside the figures are very unclear, and it is hard to understand the measurement process related to tooth angle in the matters.
C. Authors may need to provide the sufficient evidence or reference to verify that the measured tooth angle is representative and can be used as an abutment angle. Please refer to the paper I provide.
Chung, S. H., Park, Y. S., Chung, S. H., & Shon, W. J. (2014). Determination of implant position for immediate implant placement in maxillary central incisors using palatal soft tissue landmarks. International Journal of Oral & Maxillofacial Implants, 29(3), 627-633.
D. The data should be evaluated by statistical method.
E. The measured tooth or the adjacent tooth has metal scattering problems due to amalgam, which will cause a great measurement error, why not rule it out?
6. The descript of conclusion is vague. It should be presented according to the results.
Author Response

(The authors gave the same response as above.)

Reviewer 4 Report
The idea of customizing healing or final abutment to improve soft tissue seal and emergence profile is innovative ; however, well controlled clinical studies are needed to compare outcomes of such abutments with inexpensive cylindrical stock abutments.
In most cases when dental implants are placed , significant bone remodeling and adjacent teeth drifting are occurred, which alter the space available for an anatomic abutment that represent the missing tooth. Therefore, individualized or customized design likely needed for optimal aesthetic and function. Using dimension of an ideal missing tooth for healing abutment may not adequately provide the optimal emergence profile for an edentulous area that had suffered from dynamic changes of bone and adjacent teeth movements.
This study also uses linear measurements from CBCT to design 3-D objects , which may not represent accurate 3-D profile of the missing tooth.
The detailed anatomic measurements of human teeth are also available in the literature , so I am not sure the actual value of the CBCT measurements, unless , authors are referring to a specific ethnic population. From the paper , the intention is not clear.
Finally, the English language of this paper requires extensive editing and I strongly suggest the help of a professional English editor.
Author Response

(The authors gave the same response as above.)

Round 2
Reviewer 1 Report
Dear authors,
Thank you very much for your responses. Certainly, the paper has been improved. I have no comments about the manuscript.
Reviewer 3 Report
In line 103, there is a typing error, please correct it to be "implantology"
During line 260-264, the size of words is not the same, please correct it.